# An internet-delivered acceptance and commitment therapy program for anxious affect, depression, and wellbeing: A randomized, parallel, two-group, waitlist-controlled trial in a Middle Eastern sample of college students

Zahir Vally[1]*, Harshil Shah[1], Sabina-Ioana Varga[1], Widad Hassan[1], Mariam Kashakesh[1], Wafa Albreiki[1], Mai Helmy[2]

1 Department of Clinical Psychology, United Arab Emirates University, Al Ain, United Arab Emirates,
2 Department of Psychology, Sultan Qaboos University, Muscat, Sultanate of Oman

* zahir.vally@uaeu.ac.ae

## Abstract

### Background

College students during the young adult years are at elevated risk for the development of anxiety and depressive difficulties. Moreover, a preliminary body of evidence suggests that, for those who reside in Middle Eastern contexts, despite an established need, sociocultural impediments prevent active psychological help-seeking. Internet-delivered, self-directed mental health programs may hold significant promise to alleviate these difficulties in contexts where individuals would otherwise not enlist the support of a mental health practitioner.

### Method

The present study developed a bespoke, 4-module, internet-delivered program based upon acceptance and commitment therapy (ACT) principles and tested its feasibility and efficacy within the context a randomized controlled trial. A total of 129 participants were randomized to receive either the ACT program or to a waitlist control condition. Assessments of generalized anxiety, social anxiety, depressive affect, and wellbeing were administered at baseline and at post-intervention.

### Results

Analyses indicated that the intervention was efficacious in mitigating both generalized and social anxiety and in improving wellbeing.

**Data Availability Statement:** All relevant data are within the manuscript and its Supporting information files.

**Funding:** The author(s) received no specific funding for this work.

**Competing interests:** The authors have declared that no competing interests exist.

## Conclusion

These results provide preliminary evidence of the feasibility and efficacy of internet-delivered ACT in a Middle Eastern context.

## Introduction

Depressive and anxiety difficulties are some of the most commonly experienced psychological issues in the general population [1]. They are also some of the most frequently reported reasons for seeking psychological assistance amongst college-aged students [2, 3]. Given that the stage of development represented by the college years is a period of immense transition and, thus, appear to be associated with an elevated probability for the onset of psychological difficulties such as depression, anxiety, and substance use disorders [4, 5], it is unsurprising that research consistently finds high prevalence rates for these disorders among college-aged students [1, 2, 6]. This elevated prevalence is significant given its occurrence at a time of developmental significance and the commensurate psycho-emotional distress that it precipitates during this period of life transition. Research has also shown its deleterious impact on overall functioning and, in particular, academic performance [2, 7]. Therefore, the college years stand as a distinctive phase that necessitates the exploration of the efficacy of treatment approaches targeting debilitating mental health conditions during this stage of life.

Various short-term psychological interventions have demonstrated effectiveness in addressing anxiety, stress, and depression among college students. Cognitive-behavioral therapy (CBT) interventions [8–11], as well as mindfulness and acceptance-based approaches [12, 13], have produced efficacious results in college settings. However, it remains concerning that only a limited number of college students struggling with anxiety are able to access mental health support. The American College Health Association's (ACHA) National College Health Assessment highlights that, in 2020, a meagre 19.4% of the students surveyed received treatment for an anxiety difficulty, despite a substantial proportion of respondents (63%) reportedly experiencing overwhelming anxiety. A further 23.3% reported that anxiety had a deleterious impact on their academic performance [14]. These findings concur with those of preceding epidemiological surveys of college students. Findings from the ACHA survey in 2014 indicated that only 14.3% of respondents reported receiving a professional diagnosis or treatment for anxiety difficulties and this is despite a total of 54% having reported the experience of overwhelming anxiety [14]. In data derived from the National Epidemiologic Survey on Alcohol and Related Conditions, a mere 15.9% of respondents report having received psychological support for an anxiety difficulty [15].

A number of obstacles impede access to essential mental health services for college students [16]. Several educational institutions currently struggle with financial challenges, potentially hindering their ability to adequately address students' psychological needs and limiting students' access to evidence-based treatments [17]. It has been reported that some institutions face an average counselor-to-student ratio of 2,624:1, resulting in a substantial therapist shortage and inordinate waitlists [18]. Moreover, mental health stigma compounds the problem in some communities where issues of feasibility, acceptability, and cultural appropriateness deters individuals from seeking care [19]. Despite the indication that Arab individuals resident in Middle Eastern settings have become more informed about mental health, reducing the likelihood of promoting stigmatized beliefs, there is conflicting evidence suggesting that stigma remains deeply ingrained and continues to impede psychological help-seeking behaviour [20, 21].

Given the prevalence of mental health difficulties amongst the college-aged population and the demonstrated obstacles in accessing mental healthcare, particularly amongst individuals resident in Middle Eastern settings, the conventional approach of providing services primarily via face-to-face individual therapy or brief group interventions is inadequate. There exists a clear need for the development and evaluation of innovative methods to enhance access to evidence-based treatments. Internet-delivered interventions have recently emerged as a viable alternative to address this need, potentially reducing suffering and disparities in access to mental healthcare [22]. A number of psychotherapeutic programs delivered via computer or smartphones have been developed and subjected to empirical evaluation. These range from those offering psychological support by psychotherapists in real-time to programs that are designed to be entirely self-directed by the client.

Internet-delivered CBT has demonstrated effectiveness in addressing anxiety disorders and depression in both the general population [23–25] and in college samples [26, 27]. Additionally, a burgeoning evidence base in support of mindfulness and acceptance-based therapy methods has begun to emerge. Specifically, the evidence suggests that these approaches hold potential to yield reductions in depression and anxiety and improve overall quality of life across various groups of individuals [28–30], including amongst college students [31–33]. Furthermore, research has indicated that internet-delivered CBT and acceptance-based interventions that include some form of therapist or administrator support are comparable with those delivered face-to-face. Several meta-analyses in which the efficacy of face-to-face CBT was compared with that of internet-delivered CBT for anxiety and depression have demonstrated a similar pattern of results [34–36]. This is especially noteworthy given that web-based programs demand substantially less therapist time compared to face-to-face psychotherapy [37].

## Study aims and hypotheses

Based on the preceding literature, our objective was to create and assess the feasibility and efficacy of a self-directed, internet-delivered program, based on acceptance and commitment therapy (ACT) principles. This program consisted of four modules and aimed to address anxiety, depression, and wellbeing in a sample of Arab college students residing in the Middle East. A comparative analysis was conducted against a waitlist control group. Notably, this study is, to our knowledge, the first to examine the efficacy of an internet-based approach, tailored to target these outcomes among college students in a Middle Eastern context. First, in terms of feasibility, it was hypothesized that it would be possible to successfully enrol students for participation in the study. Additionally, we posited that participants engaged in the intervention program would evidence significant reductions in anxiety levels (including both generalized anxiety and social anxiety) between the pre-treatment and post-treatment assessments, in comparison to those in the waitlist condition. We also anticipated similar changes to occur in relation to depression, subjective well-being, and symptom-related disability over the treatment period. Furthermore, we hypothesized that notable shifts would occur in secondary outcome measures that were chosen to mirror the proposed mechanisms of change rooted in acceptance and commitment-based interventions. These changes encompassed a decrease in experiential avoidance and a simultaneous increase in engagement in valued actions. These changes were expected to be more pronounced in the intervention group when compared to the waitlist group.

## Materials and method

### Study design and participants

This study employed a two-group, parallel, randomized, waitlist-controlled design. The study was conducted at two sites—university campuses in the United Arab Emirates and the

Sultanate of Oman. The sampling frame comprised of a random selection of 15 classes—both undergraduate and postgraduate—drawn from across the full range of faculties and degree programs offered at the two institutions. Both countries are located in the Middle East and classified as high-income countries according to the World Bank classification. Students at the targeted institutions are principally Arab in terms of ethnicity and Muslim in relation to religion. Whilst Arabic is the native language in this region of the world, English is the medium of instruction at the targeted institutions, and participants were therefore entirely bilingual.

To be included in the trial, participants were required to be presently registered students at one of the targeted institutions and aged 18 years or above. The presence of elevated anxiety or depression was not a requirement for inclusion. Students were recruited from the targeted classrooms as follows: one of the members of the research team visited the identified classrooms during the middle of the Spring semester of the 2022/2023 academic year. Data collection commenced on 20 March 2023 and concluded on 28 June 2023. The classroom visit was arranged in coordination with, and with the permission, of the respective course instructor. The researcher introduced the study by providing basic background information, explained what participation in the study entailed, and informed students of their rights and the responsibilities of the research team. Students were then provided with a quick response code that, when scanned, linked to the baseline survey, which they were able to complete immediately if they were willing to participate in the study. Randomization was executed within 7 days of baseline data collection (see below for randomization and masking).

## Ethics

All participants provided written informed consent prior to participation. The survey first presented an informed consent form alongside background information relating to the study. The responsibilities of the research team and the rights of participants were highlighted. Participants were informed that participation in the study was entirely voluntary, that no compensation for participation would be offered and that they were free to withdraw from the study, at any point, without penalty. The contact details of the principal investigator (ZV) were provided if questions or concerns about the study and/or participation arose. The study received ethical approval for its conduct from the Social Sciences Research Ethics Committee at the United Arab Emirates University (Reference number: ERSC_2022_1866). The trial was conducted in accordance with the principles of the Declaration of Helsinki and its later amendments.

## Randomization and masking

A total of 129 participants provided informed consent and completed the baseline survey (see Fig 1. participant flow diagram). They were informed at baseline that they would receive access to the intervention program, either immediately, or after 4 weeks. Participants were then randomly allocated, by the principal investigator (ZV), using a 1:1 ratio and a minimization approach that controlled for demographic characteristics, to one of two groups, an intervention group which received access to the self-directed, online intervention immediately or to a waitlist control group. Participants assigned to the intervention group received access to the first module of the intervention program within 7 days of having completed the baseline survey. Participants assigned to the control group received an email communication also within 7 days of having completed the baseline survey. This served the purpose of communicating to control participants that access to the program would follow. Following the post-intervention assessment of all participants in week 5, all control participants then received access to the complete program, however, no follow-up assessment of these participants was completed.

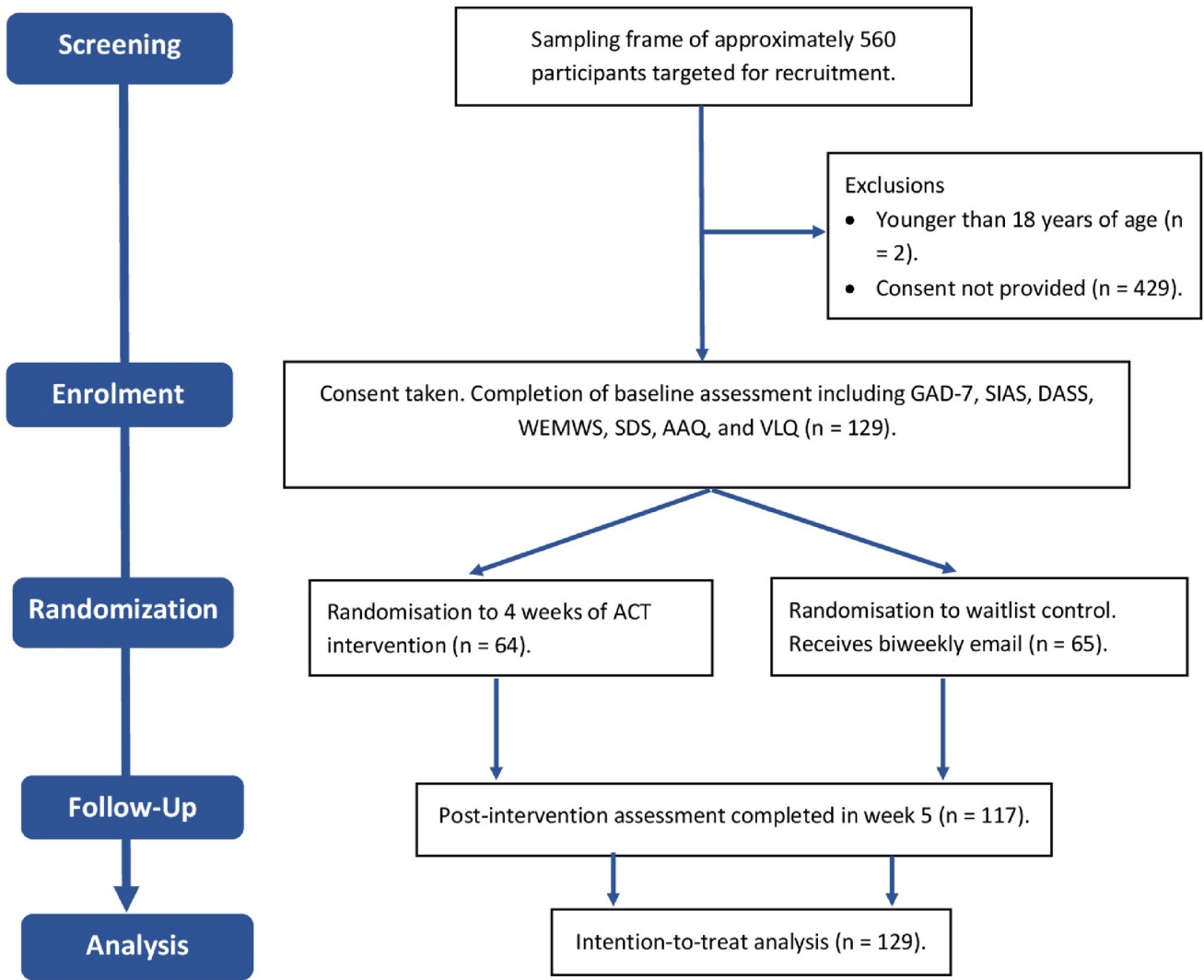

**Fig 1. Participant flow diagram illustrating flow of participants through the trial.**

## Description of the intervention program

A self-directed, four-module ACT-based program was designed and provided to students via the university's online learning platform. Participants assigned to the intervention group were provided with access to a set of PowerPoint slides with an overlayed narration in which the content for the week's module was explained. The slides contained text, images, and instructions for experiential activities. Participants were provided with access to the material of one module, per week, over a four-week period. In week five, post-intervention assessments were completed. Access to the material was provided on a weekly basis to ensure that participants worked through the material progressively as the modules were designed to incrementally build upon each other in relation to knowledge development and skills practice. The program was designed by a team of Masters-level trainees in clinical psychology under the supervision of an experienced senior clinical psychologist (ZV). The content and design of each of the four modules that comprised the program are described in more detail in the supporting information to this manuscript (S1 File).

## Assessment instruments

**Generalized Anxiety Disorder-7.** Generalized anxiety was assessed in the sample using the Generalized Anxiety Disorder-7 Scale (GAD-7) [38]. It consists of 7 Likert scale items and participants are asked to report to what extent they have been troubled by problems related to anxiety over the last two weeks. The 4-point Likert scale ranges from "Not at all" to "Nearly every day" scored as 0 to 3 respectively. The scores of all 7 items are combined to produce a total score which can range from 0 to 21. Some examples of items in the form are, "Worrying too much about different things" and "Becoming easily annoyed or irritable". A final question at the end of the scale measures the extent to which functioning in different contexts such as work, home and relationships with others has been affected by GAD using a 4-point Likers scale ranging from "Not difficult at all" to "Extremely difficult". In the present sample, internal consistency was acceptable ($\alpha = .78$).

**Social Interaction Anxiety Scale.** The 19-item Social Interaction Anxiety Scale (SIAS) was developed by Mattick and Clark [39] as a tool to measure anxiety related to general social interaction. Participants are asked to respond to what extent they agree with statements that assess the degree of fear associated with interaction with others. On a 5-point Likert scale, the responses can range from "Not at all" to "Extremely". "I worry about expressing myself in case I appear awkward" and "I am at ease meeting people at parties, etc." are some examples of the statements that appear in this scale. For the SIAS, in the present sample, internal consistency was excellent ($\alpha = .90$).

**Depression Anxiety Stress Scale.** The Depression Anxiety Stress Scale (DASS) is a 42-item questionnaire that can be utilised to measure symptoms of depression, stress, and anxiety [40]. In this study, depressive symptoms were measured in the sample using only the depression scale of the 21-item version of the DASS. The depression scale of the DASS comprises of 7 items scored on a 4-point Likert scale where respondents are asked to what extent the statements presented in the questionnaire applied to them over the past week on a 4-point Likert scale which ranges from "Never" to "Almost Always". Some representative items of this subscale include, "I felt downhearted and blue" and "I felt I wasn't worth much as a person". For this subscale of the DASS, using the present study's data, internal consistency was high ($\alpha = .83$).

**Warwick-Edinburgh Mental Well Being Scale (Short Version).** The Warwick-Edinburgh Mental Well Being Scale (WEMWBS) was developed with a large, representative sample in the UK to measure mental wellbeing using a 14-item scale with positively worded statements [41]. The shorter version (SWEMWBS) consists of 7-items and participants are asked to what degree they experienced a particular set of feelings and thoughts over the past two weeks, such as "I've been feeling useful" and "I've been feeling close to other people". Responses are scored on a 5-point Likert scale ranging from "None of the time" to "All of the time". The SWEMWBS was found to have high correlation with, and similar performance compared to the WEMWBS in a British and Norwegian Sample and a higher response rate relative to the original [42, 43]. Internal consistency was acceptable in the present study ($\alpha = .75$).

**Sheehan Disability Scale.** The Sheehan Disability Scale (SDS) is a self-report measure of the impact on functioning due to mental disorders and is widely used to assess symptom-related disability [44]. The scale consists of three 5-point Likert items ranging from "Not at all" to "Extremely" which gauge functioning in school/work, social life and family life and two items at the end which measure the number of days lost and number of days spent unproductive. Sample statements from the scale include, "The symptoms have disrupted your family life/home responsibilities" and "The symptoms have disrupted your work/schoolwork". Internal consistency was acceptable in the present study ($\alpha = .80$).

**Acceptance and Action Questionnaire—II.** Experiential Avoidance and Psychological Inflexibility were measured using the Acceptance and Action Questionnaire—II (AAQ—II), which is a 7-item scale rated on a 7-point Likert where respondents are asked to rate the extent to which the statement is true with answers ranging from "Never true" to "Always true" [45]. Some of the statements found in the AAQ-II are "I worry about not being able to control my worries and feelings" and "Emotions cause problems in my life". The AAQ-II, compared to the original AAQ was found to have superior psychometric properties that are stable in across different groups [45]. In the present sample, the AAQ-II was internally consistent ($\alpha$ = .89).

**Valued Living Questionnaire.** The Valued Living Questionnaire (VLQ) [46] is a brief, 20-item questionnaire that can be used to measure 10 valued domains of life, namely: (a) Family, (b) Intimate Relationships, (c) Parenting, (d) Friendship, (e) Work, (f) Education, (g) Recreation, (h) Spirituality, (i) Citizenship, and (j) Physical self-care. The questionnaire is divided into two parts with 10 questions each that are scored from 1 to 10 on a 10-point Likert scale. In the first part, respondents are asked to what extent they value the ten areas of life mentioned above from 1 ("not at all important") to 10 ("extremely important") and second section measures the degree to which their actions over the past week were consistent with their values ranging from 1 ("not at all consistent with my value") to 10 ("completely consistent with my value") [46]. In the present study, the VLQ produced a high Cronbach's alpha value indicating that it was internally consistent ($\alpha$ = .92).

## Data analysis

An a priori power calculation was computed using G*Power to determine the minimum sample size required using the following indices—to detect a large effect of f = .4 and $\alpha$ set at.05, it was determined that a minimum sample of 84 participants was needed to attain power of.95. The following analyses were conducted on the resultant dataset. The data were first examined to determine the extent of missing data. Mean replacement methods were used to impute missing fields for the baseline measurements. Where participants were entirely untraceable and thus no post-intervention data were available, multiple imputation methods were used to complete the dataset and an intention-to-treat analysis was used for all consequent analyses. Multiple imputation was implementation in three steps. First, missing values were estimated, multiple times, to create several complete datasets so as to account for uncertainty in the missing data. Second, each resultant complete dataset was analyzed independently. Finally, the results from these multiple analyses were then combined to produce overall estimates, adjusted for the variability between imputations.

Descriptive statistics were computed for all demographic characteristics as well as for the complete set of outcome variables. To do so, means and standard deviations were computed for the continuous variables and counts and percentages for the categorical variables. To examine the effect of the intervention program, post-intervention mean scores for each outcome measure were compared between the intervention and control groups using an analysis of covariance (ANCOVA) that included, in its computation, the associated baseline mean scores for each variable as a covariate. This method allows for the potential effect that variation in baseline scores between the two experimental groups on the post-intervention analysis may have to be controlled for [47]. This method of analysis was followed for each of the eight outcome variables examined. The results of the analyses were regarded as statistically significant where p < .05 and the associated magnitude of the effect was expressed using partial eta squared ($\eta_p^2$). In interpreting the effect size, the following rules of thumb were used: $\eta_p^2$ = .01 was regarded as small, $\eta_p^2$ = .06 was interpreted as a medium effect, and $\eta_p^2$ = .14 was regarded as a large effect size [48].

**Table 1. Demographic characteristics of sample at baseline.**

| | | Intervention (n = 64) | Control (n = 65) | $\chi^2 / t$ |
|---|---|---|---|---|
| **Mean age (years)** | | 20.72 (1.46) | 20.67 (1.57) | .156 |
| **Country of residence** | UAE | 40 (62.5%) | 42 (64.6%) | .062 |
| | Oman | 24 (37.5%) | 23 (35.4%) | |
| **Gender** | Male | 20 (31.3%) | 19 (29.2%) | .062 |
| | Female | 44 (68.8%) | 46 (70.8%) | |
| **Year of study** | 1st | 5 (7.8%) | 7 (10.8%) | 5.029 |
| | 2nd | 19 (29.7%) | 15 (23.1%) | |
| | 3rd | 18 (28.1%) | 28 (43.1%) | |
| | 4th | 20 (31.3%) | 14 (21.5%) | |
| | Postgraduate | 2 (3.1%) | 1 (1.5%) | |
| **Program of study** | Psychology | 24 (37.5%) | 25 (38.5%) | 7.157 |
| | Social work | 1 (1.6%) | 2 (3.1%) | |
| | English | 6 (9.4%) | 5 (7.7%) | |
| | Political science | 3 (4.7%) | 4 (6.2%) | |
| | Biology | 4 (6.3%) | 2 (3.1%) | |
| | Computer science | 1 (1.6%) | 1 (1.5%) | |
| | Chemistry | 2 (3.1%) | 1 (1.5%) | |
| | Nutrition | 1 (1.6%) | 3 (4.6%) | |
| | IT | 4 (6.3%) | 0 (0%) | |
| | Finance | 1 (1.6%) | 2 (3.1%) | |
| | Education | 17 (26.6%) | 20 (30.8%) | |

Data are mean and standard deviation with the accompanying t-test result for the continuous variable and counts and percentages with the accompanying $\chi^2$ value for the categorical variables.

## Results

### Participants

A total of 129 participants completed the baseline survey. A total of 82 (63.6%) were from the UAE and 47 (36.4%) from Oman. The majority of participants were female (n = 90, 69.8%). The sample's age ranged from 18 to 27 years (m = 20.69, SD = 1.53). Participants were drawn from across all four years of undergraduate study (12 were 1st years, 34 in their 2nd year, 46 were completing the 3rd year, and 34 were in their final 4th year of study). A further 3 participants were postgraduates. The sample was equally diverse in terms of program of study with participants drawn from a range of faculties and degree programs. See Table 1 for an outline of the sample's descriptive characteristics. The table illustrates that the intervention and control groups were relatively similar at baseline with respect to the demographic characteristics assessed.

Fig 1 provides a participant flow diagram. Twelve participants (9.3%) were untraceable at post-intervention, resulting in a total sample of 117 participants, however, given our decision to employ an intention-to-treat analysis, multiple imputation was used to derive post-intervention scores for these participants and thus all 129 participants were included in all analyses. Attrition was not related to group assignment, to any particular demographic variable, or to the baseline scores of any of the outcome measures.

### Baseline differences

The baseline mean scores across all eight primary outcome measures were compared between the two experimental groups (ACT versus waitlist) to determine whether pre-treatment group

differences were present. T-test comparisons revealed that no baseline differences were evident in relation to any of the examined variables (all p >.05).

## Anxiety

As can be seen from Table 2, on the GAD-7, participants in the intervention group produced comparatively lower scores than participants in the control group (post-intervention mean scores of 6.45 (SD = 3.90) versus 9.14, SD = 5.15 for the intervention and control groups respectively). This difference in self-reported anxiety, after controlling for the impact of potential baseline variation, was highly significant (F(1, 126) = 23.95, p < .001, $\eta_p^2$ = .16). Within-group pre to post-treatment change in the intervention group was equally significant (t(63) = 5.51, p < .001).

On the SIAS, participants in the intervention group produced post-intervention mean scores that were comparably lower than those produced by participants in the control group (mean scores of 28.03 (SD = 15.07) versus 34.69 (SD = 12.06) for the intervention and control groups respectively) (see Table 2). A post-intervention analysis in which the potential impact of baseline variation in SIAS scores were controlled for indicated that this difference in mean scores was statistically significant (F(1, 126) = 6.681, p = .011, $\eta_p^2$ = .05). A within-group analysis of the change in scores on the SIAS from pre to post-treatment was not statistically significant (t(63) = .98, p >.05). These results collectively suggest that the significant between-group result on this measure was the combined result of both a minimal, albeit statistically nonsignificant reduction in social anxiety in the ACT group in addition to a minimal elevation in SIAS scores in the control group. It is noteworthy that a within-group, pre to post-treatment examination of SIAS scores in the control group was statistically significant (t(63) = -1.91, p < .05).

**Table 2. Descriptive and inferential statistics for all outcome variables stratified by intervention group at pre- and post-intervention.**

| Outcome Variable | Range | Treatment Group | Pre (m, SD) | Post (m, SD) | F (sig.) | Mean Diff. (95%CI) | $\eta_p^2$ |
|---|---|---|---|---|---|---|---|
| GAD | 0–21 | Intervention | 9.34 (4.52) | 6.45 (3.90) | 23.95** | -2.62 (-3.68, -1.56) | .16 |
| | | Control | 9.45 (3.96) | 9.14 (4.15) | | | |
| SIAS | 0–80 | Intervention | 29.14 (14.35) | 28.03 (15.07) | 6.681* | -3.49 (-6.17, -.82) | .05 |
| | | Control | 33.06 (13.72) | 34.69 (12.06) | | | |
| DASS | 0–21 | Intervention | 11.78 (8.52) | 11.59 (8.24) | 20.54 | -.80 (-2.63, 1.02) | .006 |
| | | Control | 13.38 (7.71) | 13.63 (8.01) | | | |
| WEMWS | 14–70 | Intervention | 21.63 (4.98) | 24.03 (4.45) | 16.81** | 2.30 (1.19, 3.41) | .12 |
| | | Control | 22.12 (4.18) | 22.06 (4.37) | | | |
| SDS | 0–30 | Intervention | 12.45 (8.66) | 10.75 (8.27) | .434 | .52 (-1.05, 2.09) | .003 |
| | | Control | 13.05 (7.34) | 10.68 (6.92) | | | |
| AAQ | 7–49 | Intervention | 22.53 (9.22) | 19.72 (9.38) | 5.138* | -2.60 (-4.86, -.33) | .039 |
| | | Control | 23.83 (8.69) | 23.18 (8.21) | | | |
| VLQ | 10–100 | Intervention | 58.77 (23.16) | 62.31 (24.17) | 2.052 | 3.53 (-1.35, 8.40) | .016 |
| | | Control | 57.82 (19.94) | 58.01 (20.43) | | | |

Data are mean and standard deviation for each variable and for each assigned group. Result indicates F-statistic and its associated statistical significance.

**p < .001;

*p < .05.

$\eta_p^2$ = partial eta squared; GAD = Generalized Anxiety Disorder; SIAS = Social Interaction Anxiety Scale; DASS = Depression Anxiety Stress Scales; WEMWS = Warwick Edinburgh Mental Wellbeing Scale; SDS = Sheehan Disability Scale; AAQ = Acceptance and Action Questionnaire; VLQ = Valued Living Questionnaire.

## Depression

Table 2 illustrates the baseline and post-intervention scores for the DASS. This indicates that participants in the intervention group produced lower depression scores compared to their counterparts in the control group (post-intervention mean scores of 11.59 (SD = 8.24) versus 13.63 (SD = 8.01) for the intervention and control groups respectively). However, despite this difference, a post-intervention analysis indicated that this difference did not reach statistical significance (F(1, 126) = 20.543, p >.05, $\eta_p^2$ = .006).

## Wellbeing

Participants in the intervention group produced higher scores on the WEMWS compared to participants assigned to the control group at post-intervention (mean scores of 24.03 (SD = 4.45) versus 22.06 (SD = 4.37) for the intervention and control groups respectively). This mean difference at post-intervention was statistically significant when controlling for the impact of baseline scores (F(1, 126) = 16.81, p < .001, $\eta_p^2$ = .12). A further post-hoc analysis of within-group change on the WEMWS measure in the ACT group revealed a statistically significant difference from pre to post-treatment (t(63) = -4.58, p < .001).

## Disability

Self-reported scores on the SDS were relatively similar at post-intervention (see Table 2). The sample produced post-intervention mean scores of 10.75 (SD = 8.27) and 10.68 (SD = 6.92) for the intervention and control groups respectively. The consequent ANCOVA therefore was not significant (F(1, 126) = .434, p >.05, $\eta_p^2$ = .003).

## Experiential avoidance

Participants who received the intervention produced statistically significant reductions in experiential avoidance when compared to participants in the control group (post-intervention mean scores on the AAQ-II of 19.72 (SD = 9.38) versus 23.18 (SD = 8.21). The results of the ANCOVA in which these post-intervention scores were compared, and the effect of baseline scores controlled for, indicated a difference that was statistically significant (F(1, 126) = 5.138, p = .025, $\eta_p^2$ = .039). Moreover, preliminary associative analyses appeared to suggest that change in experiential avoidance over the course of the intervention (i.e., post-pre AAQ scores) were associated with generalized anxiety (r = .20, p < .05) and thus, potentially, a reduction in experiential avoidance may explain individuals' commensurate reduction in experienced anxiety, but this hypothesis should be tested within the context of further mediation and moderation analyses.

## Values-based living

Examination of the baseline and post-intervention scores on the VLQ demonstrates that participants who received the intervention produced higher overall mean scores at post-intervention compared to those assigned to the control group (mean scores of 62.31 (SD = 24.17) versus 58.01 (SD = 20.43) for the intervention and control groups respectively). While a numerical difference was evident, this was minimal and thus not statistically significant (F(1, 126) = 2.052, p >.05, $\eta_p^2$ = .016). Change in the VLQ measure between baseline and post-intervention (i.e., post-pre VLQ scores) was also inversely associated with depression (r = -.17, p < .05) and positively associated with wellbeing (r = .18, p < .05) suggesting that this may indeed act an important mechanism for change following an ACT intervention, however, as is the case for experiential avoidance, this contention should be tested in further analyses.

## Discussion

The results of this study contribute to our evolving understanding of the feasibility and efficacy of internet-delivered mental health interventions for college-aged young adults in Middle Eastern contexts. In relation to feasibility firstly, a significant proportion of the potential participant pool who were approached for participation, provided consent and completed the baseline survey, indicating that individuals in this context may indeed be interested in a mental health intervention of this sort. Approximately 23% of the sampling frame were included in the trial. This is in keeping with the typical rates of participant conversion for recruitment into most randomized trials [49]. However, despite strong interest, some attrition was evident. Almost 10% of the participants who were initially recruited and randomized failed to complete the post-intervention assessments which we define as drop-out. In most randomized trials of psychotherapy interventions some degree of drop-out is expected, and indeed typically much larger rates have been reported [24], especially for studies in which the interventions have been delivered via the internet [35].

Participant engagement and rates of program completion are additional issues that appear to be pronounced in trials of internet-delivered programs and are of integral importance given that varying degrees of engagement with the intervention may draw into question whether participants indeed received the intervention under investigation. In the present trial, we were unfortunately unable to measure reliable metrices of engagement and completion. Doing so reliably is often problematized by the varying potential ways of assessing the constructs of engagement, adherence, and completion, thus complicating comparison between studies. Indeed, in future iterations of the present trial, at post-intervention, participants could be requested to provide data on the number of modules that they completed, the approximate time spent on each module, and the number of module-related assignments that were successfully completed. These may all provide potential indications as to the extent to which participants were engaged in the program and doing so using an objective method would be most reliable.

In relation to the assessment of the efficacy of the program, analyses of the results for all variables produced mixed findings. Participants who received the intervention produced appreciable and statistically significant reductions in both generalised anxiety and social anxiety as well as improved self-reported wellbeing between pre- and post-intervention. However, there were no significant changes to our measure of depressive affect (DASS). In relation to secondary outcomes, analysis of the SDS measure was not significant across time-points and so too was the analysis of engagement in valued living. Of note, however, on the VLQ, participants assigned to the intervention group reported greater engagement in values-driven behaviours across time-points (mean scores of 58.77 at baseline and 62.31 at post-intervention) but this difference was not statistically significant. It may be likely that the 4-week duration between assessment points is not a sufficient timeframe with which to reliably observe a noticeable and significant impact on this outcome. This is a sound contention as individuals may require time to interpret their values, following engagement in values-clarification activities, and then to implement them into clearly defined and tangible domains in their lives. Thus, additional follow-up points may have produced significant results, but these were not feasible in the present study. Analysis of the AAQ indicated that the intervention produced reductions in experiential avoidance and therefore, it is likely this construct that the intervention indirectly targeted which, in turn, produced efficacious change to the principal outcome variables but this contention could be tested in a further mediational study of change processes.

The consistent improvements in the intervention group on both measures of anxiety, the generalized and social anxiety variants, as well as wellbeing are in concurrence with the preceding literature that has collectively demonstrated that internet-delivered mental health programs based upon ACT and mindfulness principles are effective in targeting these outcomes and these findings appear to occur uniformly across varying samples of participants [26, 27, 31, 34]. We can now further conclude, on the basis of this study's results, that these conclusions can be extended to college-aged participants in Middle Eastern contexts. The effect sizes across the assessed variables varied somewhat but are generally consistent with those evidenced in previous studies of internet-delivered interventions [16, 24, 25]. These were generally in the small to moderate range with the exception of the GAD-7 which produced an effect of large magnitude.

We did not assess whether the effects at post-intervention persisted across time as follow-up assessments were not feasible at this time. However, these would be essential to ascertain whether the effects accrued from such a program of intervention are able to produce long-lasting change. Additionally, exploration of the use of booster sessions could be a worthwhile endeavour alongside assessments of their efficacy. Our results already appear to suggest that implementation of behaviours consistent with individuals' values can be catalysed by such a program, but a longitudinal design would be needed to determine whether they persist and accrue in magnitude across time.

With the consideration in mind that participant engagement and rates of completion have been demonstrated to be lower for internet-delivered interventions, in comparison to those provided face-to-face, we deliberately designed the program to be brief (i.e., only 4 sessions) and each module to be succinct and measured in the proportion of information imparted (i.e., requiring, on average, no more than 30 minutes to work through a single module). Despite this, further measurements of the extent to which participants accessed the material (if the shared material were opened or not), the amount of time spent on each module, and information pertaining to the specific activities engaged in when the files were accessed (e.g., whether all information was listened to, whether activities were completed, whether participants multi-tasked and completed other computer-based tasks while listening to the recorded module) would all be valuable data to make a reliable determination of engagement and completion. The information contained in the module presentations were also deliberately designed to be less reliant on literacy capacity to account for varying levels of language proficiency. The presented information made liberal use of written text, employed images and videos, and the narrated audio so as to be less reliant on written engagement from participants. Studies that require participants to submit written communication in response to practice assignments may encounter issues related to linguistic proficiency.

The results of this study suggest that the program successfully engaged students who would have benefitted from some form of intervention but who may not have volitionally sought contact with mental health services. On the GAD-7, participants across both groups produced baseline scores indicative of mild to moderate anxiety, whilst on the SIAS, mean scores were close to the threshold indicative of social phobia. These results further underscore the importance of developing intervention programs that overcome impediments to help-seeking such as stigma and the availability of face-to-face services and have the capacity to access students who might otherwise not have engaged in psychotherapy.

This study makes a significant contribution to the literature by providing evidence of the efficacy of a brief, internet-delivered intervention for anxiety and wellbeing and is the first to demonstrate the applicability of such a form of intervention in a Middle Eastern context. This is significant given the following considerations. First, the resources that face-to-face psychotherapies demand are immense and, thus, a demonstration of efficacy of a brief intervention

delivered entirely via the internet with minimal input from a clinical team holds substantial benefits for the continued financial viability of mental health services. Moreover, in sociocultural contexts where stigma continues to impede help-seeking engagement, especially during the vulnerable developmental stage of young adulthood, a minimally intrusive intervention that is almost entirely self-directed is an especially valuable addition to the evidence base. This study also adds to the accumulating evidence in support of the cross-cultural relevance of interventions informed by ACT and mindfulness principles given that the vast majority of internet-delivered trials have been conducted with principally White, European, or North American samples.

Whilst this study provides preliminary evidence of efficacy in this location, future studies should examine the impact on efficacy that interventions of varying duration and frequency may have. A number of various iterations may be tested, for example, programs that are longer in duration (more than 4 modules), multi-session modules rather than a single session per week, compulsory rather than optional completion and submission of homework tasks, or those that involve more overt therapist input (e.g., receipt of feedback following the therapist's review of the submitted homework tasks).

Furthermore, it would be wise for future studies of this sort to include, in its design, assessment of detailed accounts of participant attrition, adherence to the intervention, and a determination of the extent to which the study materials and homework tasks are completed, and these should be accompanied by clearly operationalized definitions for these variables. Studies implemented in this region of the world should also investigate the degree to which the strategies encompassed in these programs are culturally appropriate. Whilst the recruitment and completion rates for the participants in the present study suggest that the tenets inherent in the ACT modality of intervention and the mode of implementation using the internet were acceptable to participants on this occasion, a more specific and therefore reliable marker of acceptability should be included in trials conducted with participants of Arab and/or resident in the Middle East.

An additional limitation that is acknowledged is the lack of multiplicity adjustment to the multiple secondary comparative analyses. While we acknowledge that it is becoming increasingly common to do so, particularly within the context of further subsequent analyses of trial datasets, our assessment of the secondary outcomes in this trial are inherently exploratory in nature and thus should be interpreted with caution.

A further consideration in relation to the study's design which potentially have impacted the efficacious outcome, is the choice of comparator. In this case, a waitlist control was employed, as this was the most feasible comparison condition with resources available to the research team. However, it should be noted that there is evidence that trials that have deployed active controls in which participants have received equivalent placebo attention or a comparator psychotherapeutic intervention, often fail to produce statistically significant improvement to mental health and, conversely, this is not always the case where non-active waitlist controls are implemented [50]. Thus, a further variation of the present study might consider the use of an active control, for example, ACT versus cognitive restructuring and/or behavioural activation, the alternate gold standard psychotherapeutic interventions for depressive and anxious affect.

Finally, a notable limitation of this study is its primary use of self-report measures. We acknowledge that structured diagnostic tools for assessing these psychological outcomes would be preferable, given their greater rigor and reliability. Additionally, self-report measures are prone to challenges such as recall accuracy and social desirability bias which may be remedied with the use of objective, structured measures.

## Conclusion

In conclusion, this brief, online, principally self-directed ACT intervention, designed specifically for implementation in this trial, was demonstrated to be feasible and efficacious in alleviating anxiety and promoting wellbeing among a group of college students of Arab descent resident in the Middle East. This is the first investigation of a short-term internet-delivered program designed specifically to address anxiety in the college student population in a Middle Eastern context. The findings of this study provide preliminary evidence suggestive of its efficacy and therefore further exploration is warranted.

## Supporting information

**S1 File. Description of the program's content.** Complete description of the content for each module contained in the program.
(DOCX)

**S2 File. Complete dataset for the study.** Dataset on which the study's analyses are based.
(XLSX)

## Author Contributions

**Conceptualization:** Zahir Vally.

**Data curation:** Harshil Shah, Sabina-Ioana Varga, Widad Hassan, Mariam Kashakesh, Wafa Albreiki, Mai Helmy.

**Formal analysis:** Zahir Vally.

**Investigation:** Zahir Vally, Harshil Shah, Sabina-Ioana Varga, Mariam Kashakesh, Wafa Albreiki.

**Methodology:** Zahir Vally, Harshil Shah, Sabina-Ioana Varga, Widad Hassan.

**Project administration:** Zahir Vally, Widad Hassan, Mariam Kashakesh, Wafa Albreiki.

**Resources:** Mai Helmy.

**Software:** Zahir Vally.

**Supervision:** Zahir Vally.

**Writing – original draft:** Zahir Vally.

**Writing – review & editing:** Harshil Shah, Sabina-Ioana Varga, Widad Hassan, Mariam Kashakesh, Wafa Albreiki, Mai Helmy.

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
