## [Decision Letter · Decision Letter 0]

12 Apr 2024

PONE-D-24-06315An internet-delivered acceptance and commitment therapy program for anxious affect, depression, and wellbeing: A randomized, parallel, two-group, waitlist-controlled trial in a Middle Eastern sample of college studentsPLOS ONE

Dear Dr. Vally,

Thank you for submitting your manuscript to PLOS ONE. After careful consideration, we feel that it has merit but does not fully meet PLOS ONE’s publication criteria as it currently stands. Therefore, we invite you to submit a revised version of the manuscript that addresses the points raised during the review process.

Be sure to:to discuss the rationale for choosing ACT as opposed to other third-wave therapeutic options including Transdiagnostic CBT, MBCT, or DBTHow is ACT a better choice than some of the other treatments?Who administered the structured interviews and what was their training? and Was the ACT intervention adapted culturally?  How did you control the training for the therapist? and what kind of training did they have?reply to reviewers comments.==============================

We look forward to receiving your revised manuscript.

Kind regards,

Fadwa Alhalaiqa

Academic Editor

PLOS ONE

Journal Requirements:

Reviewers' comments:

Reviewer's Responses to Questions

**Comments to the Author**

1. Is the manuscript technically sound, and do the data support the conclusions?

Reviewer #1: Yes

Reviewer #2: Partly

2. Has the statistical analysis been performed appropriately and rigorously? 

Reviewer #1: Yes

Reviewer #2: Yes

3. Have the authors made all data underlying the findings in their manuscript fully available?

Reviewer #1: Yes

Reviewer #2: Yes

4. Is the manuscript presented in an intelligible fashion and written in standard English?

Reviewer #1: Yes

Reviewer #2: Yes

5. Review Comments to the Author

Reviewer #1: Thank you for allowing me to review this well-written and interesting manuscript. I had a few thoughts regarding the study design while reading it, but most of these were discussed such as actual treatment engagement, follow up over a longer time period and such.

Overall I find this manuscript to be very well worked through, however, sometimes uneccessary detailed. I think the manuscript would benefit from being somewhat shortened, as well as use a less complicated language.

I enjoyed being able to gain insight to the content of each module of the treatment program, but maybe this could be condensed in a figure or as appendix. Regarding assessment measures the assessment scales used are well known and well used and readability would benefit from describing these in less detail.

When it comes to analysis of data I personally would have chosen a mixed-effects model over the ANCOVA model for this study as in the ANCOVA model the intervention effect, being adjusted for the baseline value as a covariate, is scaled down by a factor equal to the intraclass correlation of the outcome variable. In the mixed-effects model, lacking the covariate representation of the baseline (but including the baseline information as a separate observation in the data set) the intervention effect is estimated in its natural metric, not scaled down. For future studies, reconsider the use of statistic method.

Reviewer #2: Introduction

1. In general, the Introduction tends ramble and often presents unnecessary detail and arguments. The paper would benefit from a general shortening and tightening.

2. The research carried out in relation to a high-income Middle Eastern culture should be mentioned in the introduction. However, in the absence of research work in this area, it is better to refer to this issue in the context of the statement of problem.

3. The rationale for choosing ACT as opposed to other third-wave therapeutic options including Transdiagnostic CBT, MBCT, or DBT needs to be clarified. How is ACT a better choice than some of the other treatments?

Method

4. Who administered the structured interviews and what was their training?

5. Did the authors examine whether these children have been receiving medical interventions at the same time?

6. Was the ACT intervention adapted culturally? If yes, some examples of the cultural adaptations would be helpful to add the context.

7. How did you control the training for the therapist? and what kind of training did they have?

8. The treatment adherence measures do not use a standardized and independent measure and this should be noted at a limitation.

Results

9. The APA format should be used for the tables.

Discussion

10. Discussion about the ability of the ACT intervention to change anxious and depressive symptoms is very speculative without first addressing that these changes could be due to regulating emotions.

11. In the limitations section, the authors should discuss the drawbacks of using self-report measures as the only outcome.

12. I suggest that more discussion should be given to the clinical implications of this study on children with tic disorders in the conclusion section.

Few Small Issues

There are a couple small typos in text worth fixing. There were also quite a few writing problems in the manuscript, and a general reread and corrections are needed.

Some of the statistical symbols should be italicized.

References must be double spaced.

Finally, perhaps the most important aspect of this study - that it was (presumably) conducted in a Middle Easter context - is left completely out. This point deserves added consideration and perhaps ought to be a major focus of the paper.

6. PLOS authors have the option to publish the peer review history of their article (what does this mean?). If published, this will include your full peer review and any attached files.

Reviewer #1: **Yes: **Charlotte Gentili, PhD, Lic Psychologist

Reviewer #2: No

---

## [Author Response · Author response to Decision Letter 0]

12 Jun 2024

Response to reviewers

Reviewer 1

Author responses.

- The Intro section has been condensed, substantially, by omitting info which we believe to be interesting but perhaps not essential. Hopefully, this aids readability. 

- The per-module description of content has been moved to an appendix, also in an attempt to reduce text and can be referred to by readers if they are interested to gain insight into the details thereof. 

- You suggestion of an alternate analytical method is interesting and useful. In future studies, we will definitely consider this method instead.

Reviewer 2

Comments.

1. In general, the Introduction tends ramble and often presents unnecessary detail and arguments. The paper would benefit from a general shortening and tightening.

2. The research carried out in relation to a high-income Middle Eastern culture should be mentioned in the introduction. However, in the absence of research work in this area, it is better to refer to this issue in the context of the statement of problem.

3. The rationale for choosing ACT as opposed to other third-wave therapeutic options including Transdiagnostic CBT, MBCT, or DBT needs to be clarified. How is ACT a better choice than some of the other treatments?

Response to comments 1-3. 

The Intro section has been condensed, substantially, by omitting info which we believe to be interesting but perhaps not essential. We hope this substantial reduction aids readability. An explication of the rationale in selecting ACT as the identified intervention approach, in comparison to the other available alternates, is provided (and suitable references in substantiation of this decision provided). 

4. Who administered the structured interviews and what was their training?

Structured interviews were not administered. The assessment instruments used in this study were self-report measures, all of which are described as such in the method. In fact, as reviewer 1 suggests, they are all well-known self-report measures and very commonly used. Thus, the comment relating to requisite training in respect of their administration is rendered moot. 

5. Did the authors examine whether these children have been receiving medical interventions at the same time?

First, the sample comprised young adults (those aged 18 years and above), not children. This is described very explicitly in the study’s method. 

Second, we did not inquire as to whether participants were concurrently receiving any simultaneous ‘medical’ or psychological intervention. We presume that the reviewer’s question intimates that this may have served as a potential confounding factor impacting the obtained effect of the intervention. This is precisely the kind of issue that randomization seeks to address. Randomised controlled designs, such as the kind employed in the present study, presumes that any potential factor that may likely impact the obtained effect of the assessed intervention would likely be distributed relatively equivalently across experimental groups. 

6. Was the ACT intervention adapted culturally? If yes, some examples of the cultural adaptations would be helpful to add the context.

We do not make the claim that the intervention was culturally adapted. Indeed, this may be a beneficial mode of inquiry for future studies in this context and, as such, we make this recommendation in the Discussion (‘Studies implemented in this region of the world should also investigate the degree to which the strategies encompassed in these programs are culturally appropriate’). 

7. How did you control the training for the therapist? and what kind of training did they have?

The intervention assessed in this study was entirely self-directed by participants. No face-to-face sessions occurred. Nor did participants have any direct contact with a clinician. The self-directed design of this intervention is very explicitly described, at length, in the study’s method and repeatedly referred to as ‘self-directed’ throughout the entire manuscript. Therefore, any query regarding the training credentials, intervention fidelity, or therapist supervision protocol – as would be essential for a therapist-administered intervention – is null and void. However, the profile of the team of therapists who designed the intervention is described (in the section listed ‘Description of the intervention program’).

8. The treatment adherence measures do not use a standardized and independent measure and this should be noted at a limitation.

Indeed, we agree that this is a limitation of our study. Therefore, this issue is clearly noted as a limitation and recommendations in this regard have been made (‘Indeed, in future iterations of the present trial, at post-intervention, participants could be requested to provide data on the number of modules that they completed, the approximate time spent on each module, and the number of module-related assignments that were successfully completed. These may all provide potential indications as to the extent to which participants were engaged in the program and doing so using an objective method would be most reliable.’). Further, comprehensive, discussion is provided in paragraph 2 (p.22) of the additional markers that could be included as proxy measurements for adherence. 

9. The APA format should be used for the tables.

Tables have been prepared in accordance with the author guidelines for this journal.

11. In the limitations section, the authors should discuss the drawbacks of using self-report measures as the only outcome.

As you suggest, we have now included a paragraph describing the limiting nature of self-report measures.

12. I suggest that more discussion should be given to the clinical implications of this study on children with tic disorders in the conclusion section.

This is a perplexing suggestion to make. Discussion of literature relating to children (i.e., an entirely different sample to that of the present study) and to tic disorders (i.e., a psychopathological construct that is not related, not even tenuously, to the content of this paper) would be nonsensical and inappropriate. 

13. There are a couple small typos in text worth fixing. There were also quite a few writing problems in the manuscript, and a general reread and corrections are needed.

The entire manuscript has been reviewed, extensively, by all authors, all of whom are native English speakers.

14. Some of the statistical symbols should be italicized.

Done.

15. References must be double spaced.

Done.

16. Finally, perhaps the most important aspect of this study - that it was (presumably) conducted in a Middle Easter context - is left completely out. This point deserves added consideration and perhaps ought to be a major focus of the paper.

Whilst this study was conducted in a very specific location, this was not a culturally-adapted intervention. This is essentially a proof-of-concept study – a pilot study to attempt to establish whether this modality of intervention (i.e., one that is delivered online and self-directed, based on ACT principles) is acceptable and efficacious. These issues are discussed in relation to the context in paragraph 2 (p.23) as follows:

(This study makes a significant contribution to the literature by providing evidence of the efficacy of a brief, internet-delivered intervention for anxiety and wellbeing and is the first to demonstrate the applicability of such a form of intervention in a Middle Eastern context. This is significant given the following considerations. First, the resources that face-to-face psychotherapies demand are immense and, thus, a demonstration of efficacy of a brief intervention delivered entirely via the internet with minimal input from a clinical team holds substantial benefits for the continued financial viability of mental health services. Moreover, in sociocultural contexts where stigma continues to impede help-seeking engagement, especially during the vulnerable developmental stage of young adulthood, a minimally intrusive intervention that is almost entirely self-directed is an especially valuable addition to the evidence base. This study also adds to the accumulating evidence in support of the cross-cultural relevance of interventions informed by ACT and mindfulness principles given that the vast majority of internet-delivered trials have been conducted with principally White, European, or North American samples.)

---

## [Decision Letter · Decision Letter 1]

20 Aug 2024

PONE-D-24-06315R1An internet-delivered acceptance and commitment therapy program for anxious affect, depression, and wellbeing: A randomized, parallel, two-group, waitlist-controlled trial in a Middle Eastern sample of college studentsPLOS ONE

Dear Dr. Vally,

Thank you for submitting your manuscript to PLOS ONE. After careful consideration, we feel that it has merit but does not fully meet PLOS ONE’s publication criteria as it currently stands. Therefore, we invite you to submit a revised version of the manuscript that addresses the points raised during the review process.

 Be sure to:You respond to reviewer feedback carefully Meet the publication criteria of PLOS ONE==============================

We look forward to receiving your revised manuscript.

Kind regards,

Fadwa Alhalaiqa

Academic Editor

PLOS ONE

Journal Requirements:

Reviewers' comments:

Reviewer's Responses to Questions

**Comments to the Author**

1. If the authors have adequately addressed your comments raised in a previous round of review and you feel that this manuscript is now acceptable for publication, you may indicate that here to bypass the “Comments to the Author” section, enter your conflict of interest statement in the “Confidential to Editor” section, and submit your "Accept" recommendation.

Reviewer #3: All comments have been addressed

2. Is the manuscript technically sound, and do the data support the conclusions?

Reviewer #3: Yes

3. Has the statistical analysis been performed appropriately and rigorously? 

Reviewer #3: Yes

4. Have the authors made all data underlying the findings in their manuscript fully available?

Reviewer #3: Yes

5. Is the manuscript presented in an intelligible fashion and written in standard English?

Reviewer #3: Yes

6. Review Comments to the Author

Reviewer #3: This is an interesting randomized, waitlist-controlled trial intended to assess the feasibility and efficacy of an internet-delivered acceptance and commitment therapy (ACT) program in a sample of students residing in the Middle East. Few studies have explored the applicability of ACT to non-clinical populations in this part of the world, and as such, this study is a welcome addition to the literature. Further, the online medium of the intervention has important implications, given its financial viability and potential to be rolled out (inter)nationally in student samples. The high rate of recruitment and low attrition are strengths. The manuscript has previously been reviewed and improved as a result (particularly streamlining the Introduction). There are a small number of (mostly minor) points (below), particularly concerning elaborating on the Results - once addressed, the manuscript could be published in PLOS One.

1. Methods p.12. The authors note that ‘GAD can be identified with a cut-off point of 10 and other cut-off points of 5, 10, and 15 can be interpreted as having mild, moderate, and severe levels of anxiety’. But the number of participants in each group scoring in the moderate-to-sever range is not detailed in the Results (although the mean values of each group are approaching 10). If there is no space to include, then I suggest removing this statement from the Methods. Alternatively, providing the baseline frequency (%) of participants scoring above (cut-off) thresholds in outcome measures where such thresholds exist (e.g., GAD, SIAS) would be helpful. Further consideration could be given to detailing the number of students who make reliable (and clinically significant) changes on these measures.

2. Methods p.15. Was a power calculation administered prior to the study to determine appropriate sample size? This would appear prudent even if the study is preliminary given formal hypothesis testing was administered.

3. Results p.17 top para. It is stated that ‘multiple imputation was used to derive post-intervention scores…’ for the 12 participants without post-treatment data. Can more detail about this be (briefly) provided?

4. Results p.17 top para. Was attrition related to score on any pre-treatment outcome measure?

5. Results Table 1. Small point but please add percentage values to the n values for Year of study and Program of study (appreciate that the n values are small for some study programs).

6. Results Table 2. Small point but in Table 2 please include in parentheses the possible score range of the measure, for example, ‘GAD-7 (0-21)’.

7. Results Table 2. Were there any differences at pre-intervention on outcome measures between ACT and control groups (I’m aware that this was controlled for in between-group comparisons in any case)? It would appear not, but useful to confirm, even if in a single sentence.

8. Results. Table 2. It would be helpful to provide the adjusted between-group post-treatment differences and surrounding confidence intervals (from estimated marginal means of ANCOVA) so readers can better appreciate the magnitude of observed differences (appreciate a measure of effect size has been provided), especially in view of the pre-treatment scores not always numerically equal between groups. Also, the mean (SD) values and ANCOVA output for measures are clearly shown in the table but these values are repeated in the accompanying text.

9. Results. pp.17-19. Did data distributions of outcome measures in each group approximate a Gaussian distribution? I ask because frequently in non-clinical populations measures of anxiety and depression can yield skewed data. If this is the case, then analyses should consider bootstrapping to estimate confidence intervals and associated p values.

10. Results. pp.17-19. On measures where significant post-treatment differences occurred, post-hoc analyses confirming significant within-group pre-to-post-treatment change in the ACT group would be helpful (to establish that between-group differences have not occurred in part due to worsening function in the control group).

11. Results p.19 top para. Trials of ACT often consider experiential avoidance / psychological flexibility as a process rather than outcome measure. Was there any suggestion that the degree of change on this measure was associated with (adjusted) post-treatment scores on outcome measures (or change on outcome measures)? I appreciate that mediational analysis is beyond the scope of the study (as alluded to in the Discussion) but simpler associative analyses could nevertheless be administered.

12. Discussion, Limitations pp.23-24. Was there any correction to multiple comparisons administered of secondary outcome measures on the trial? If not, then please add a statement in the limitations of the Discussion acknowledging this.

13. Discussion. I appreciate that the authors have clearly noted the lack of longer-term follow-up and absence of measurement of treatment adherence / engagement level for ACT was problematic. But another important issue is the comparison with a waitlist control rather than an active control. For example, online mindfulness-based interventions often are efficacious treatments for improving university students’ mental health when compared to non-active controls post-intervention, but not when compared to active controls (e.g., placebo attention controls containing non-specific therapeutic components) or other treatments (Alrashdi et al., 2024). Brief consideration of this would strengthen the Discussion.

References

Alrashdi DH, Chen KK, Meyer C, Gould RL. A systematic review and meta-analysis of online mindfulness-based interventions for university students: an examination of psychological distress and well-being, and attrition rates. Journal of Technology in Behavioral Science. 2024 Jun;9(2):211-23.

7. PLOS authors have the option to publish the peer review history of their article (what does this mean?). If published, this will include your full peer review and any attached files.

Reviewer #3: No

---

## [Author Response · Author response to Decision Letter 1]

6 Oct 2024

Response to reviewers 

PONE-D-24-06315R1. An internet-delivered acceptance and commitment therapy program for anxious affect, depression, and wellbeing: A randomized, parallel, two-group, waitlist-controlled trial in a Middle Eastern sample of college students

1. Given that, not all the measures employed in this study make provision for the use of cut-off scores, and indeed we have not used the cut-off scores in our principal analyses, this sentence describing the cut-off scores for the GAD-7 has been removed from the Method. Thus, no further analyses using them have been conducted or reported.

2. A power calculation was indeed computed. This is now included in the manuscript as a separate subheading in the Materials and Method section.

3. The method employed to execute multiple imputation is now described in a few additional sentences.

4. A sentence has been added stating that attrition was not related to baseline scores on any of the measures. P17, para 1.

5. Percentage values have been added – table 2.

6. Ranges for each instrument have been added to table 2.

7. No between-group differences at pre-intervention were evident. An additional subheading has been added to the Results describing the results of these comparative baseline analyses. 

8. The adjusted between-group post-treatment differences and surrounding confidence intervals have been added to table 2. 

9. We made the determination based on an examination of the data distributions that data did not significant deviate from normal and this proceeded with parametric analyses without a consideration of further bootstrapping. 

10. Further post-hoc analyses were computed for all the primary variables which emerged as statistically significant in the initial analyses – for the GAD, SIAS, and WEMWS. We have added textual descriptions of these pre- to post-treatment changes for the treatment group specifically to the paragraph that describes the results for each respective variable. 

11. We have added to each results paragraph that describes the primary results for each of the process variables – AAQ and VLQ – the results of preliminary associative analyses that are suggestive of the associations between the outcome and process variables (ie., Between AAQ and GAD, and between VLQ and depression and wellbeing). Pages 19 and 20

12. This limitation has been acknowledged and explained as suggested by the reviewer. page 25

13. We have added a paragraph that now acknowledges this aspect of the study’s design – its choice of a waitlist control, and its implications are further discussed. Pages 25 and 26.

---

## [Editor Report · Decision Letter 2]

22 Oct 2024

An internet-delivered acceptance and commitment therapy program for anxious affect, depression, and wellbeing: A randomized, parallel, two-group, waitlist-controlled trial in a Middle Eastern sample of college students

PONE-D-24-06315R2

Dear Dr. Zahir Vally,

We’re pleased to inform you that your manuscript has been judged scientifically suitable for publication and will be formally accepted for publication once it meets all outstanding technical requirements.

Kind regards,

Fadwa Alhalaiqa

Academic Editor

PLOS ONE
---

## [Editor Report · Acceptance letter]

24 Oct 2024

PONE-D-24-06315R2 

PLOS ONE

Dear Dr. Vally, 

I'm pleased to inform you that your manuscript has been deemed suitable for publication in PLOS ONE. Congratulations! Your manuscript is now being handed over to our production team.

Kind regards, 

on behalf of

Pro Fadwa Alhalaiqa 

Academic Editor

PLOS ONE